# HiTCO: High-Fidelity Memory and Combined-Objective Training for Dynamic Link Prediction

## ABSTRACT

Temporal graph link prediction is a critical task, yet existing methods face a trade-off between the computational efficiency of memory-based models and the expressive power of graph-based approaches. This contradiction is sharply highlighted by the introduction of large-scale, realistic benchmarks like the Temporal Graph Benchmark (TGB), which reveals the scalability limitations of many expressive models. To address this challenge, we introduce HiTCO (High-Fidelity Temporal Representation with Combined-Objective Training), a novel architecture that bridges this gap. HiTCO integrates a high-fidelity memory module, featuring Gated Recurrent Unit (GRU) based updates with principled gradient management, and a deep Multi-Layer Perceptron (MLP) prediction head that learns complex, non-linear interaction patterns between nodes. The model is trained with a novel combined-objective loss function that synergistically optimizes for both ranking accuracy and classification confidence, directly aligning the training process with modern evaluation protocols. Extensive experiments demonstrate that HiTCO achieves new state-of-the-art performance on several challenging datasets from the TGB suite, significantly outperforming a wide range of memory-based, graph-based, and hybrid baselines. Our work shows that a carefully designed, memory-centric architecture pairing with a powerful predictive head and a tailored training objective, can achieve superior expressiveness without sacrificing the scalability essential for real-world temporal graphs.

## 1 INTRODUCTION

Temporal graphs are powerful and ubiquitous structures for modeling dynamic systems where entities and their relationships evolve over time. They find applications in diverse domains, including social networks, financial transaction systems, transportation networks, and recommendation engines Cai et al. (2022); Kazemi et al. (2020). A fundamental task on these graphs is dynamic link prediction, which aims to forecast future interactions between entities. The ability to accurately predict future links is critical for applications such as anticipating user behavior in e-commerce, forecasting events in social media, and identifying potential fraudulent activities.

The research landscape for temporal graph representation learning is largely characterized by a dichotomy between two dominant paradigms: memory-based and graph-based models. Memory-based models, such as DyRep Trivedi et al. (2019) and the widely-adopted Temporal Graph Network (TGN) Xu et al. (2020b), maintain a memory state for each node that is updated sequentially using a recurrent mechanism as new events occur. Their primary strength lies in their computational efficiency and scalabilityZhou et al. (2022) by processing events in a streaming fashion in order to handle the massive scale of modern datasets. However, their reliance on node-level representations and simple aggregation functions limits their capacity to capture complex, higher-order structural patterns that are often crucial for link prediction Huang et al. (2023).

Another type of temporal graph representation learning is graph-based models, such as Temporal Graph Attention Network (TGAT) Xu et al. (2020a), Causal Anonymous Walks (CAWN) Wang et al. (2021b), and Neighborhood-Aware Temporal Network (NAT) Luo & Li (2022). These methods achieve high expressiveness by constructing and performing message passing over local temporal-topological subgraphs for each prediction query. This allows them to explicitly model the intricate interplay between neighboring nodes and their interaction timings. Despite their impressive perfor-

mance on smaller graphs, these models suffer from a critical weakness: prohibitive computational and memory costs. The process of sampling neighborhoods and executing graph convolutions for every event becomes intractable on large-scale graphs, a limitation starkly demonstrated by their failure to run on the larger datasets within the Temporal Graph Benchmark (TGB) Huang et al. (2023).

The introduction of the TGB marked a pivotal moment for the field Huang et al. (2023). With its large-scale datasets, diverse domains and realistic evaluation protocols, TGB treats link prediction as a ranking task evaluated by Mean Reciprocal Rank (MRR) with a mix of challenging negative samples. Also, TGB provides a more rigorous and less "over-optimistic" assessment of model capabilities. The benchmark's results reveal that many classical models struggle with either scalability or robustness under these realistic conditions, which exposes the urgent need for methods that are both expressive and efficient Huang et al. (2023).

This need has catalyzed the development of hybrid approaches that seek to combine the strengths of both paradigms. A notable example is the Temporal Neural Common Neighbor model, which augments a memory-based backbone with an explicit, heuristic-based module for extracting pairwise structural features like common neighbors Huang et al. (2023). While effective, this approach relies on a pre-defined structural heuristic that may not be optimal for all datasets or dynamics.

In this paper, we propose HiTCO, a novel architecture that addresses the efficiency-expressiveness trade-off in a more principled and general manner. HiTCO advances the memory-based paradigm by integrating a high-fidelity memory module with a powerful, deep predictive head and a tailored training objective, designed to learn optimal pairwise features directly from data rather than relying on hand-crafted heuristics. Our contributions are as follows:

- We introduce HiTCO, a novel and scalable architecture for temporal link prediction that integrates a high-fidelity GRU-based memory module with a deep MLP predictive head to learn rich and non-linear pairwise representations.

- We propose a combined-objective loss function that synergistically optimizes for both ranking accuracy via a margin-based loss and classification confidence via binary cross-entropy aligning the model's training directly with the demands of modern ranking-based evaluation protocols.

- We conduct extensive empirical validation on five large-scale datasets from the TGB. Our results show that HiTCO establishes a new state-of-the-art, outperforming a comprehensive suite of heuristic, memory-based, graph-based, and hybrid baselines in both effectiveness and efficiency.

- We perform a thorough analysis and ablation study that systematically dissects HiTCO's architecture providing clear evidence for the contribution of each novel component to its overall performance.

## 2 RELATED WORKS

Our work is situated within the broader context of representation learning on temporal graphs. The field can be broadly categorized into several key research directions.

**Memory-Based TGNNs:** This line of work focuses on efficiency and scalability by maintaining a memory state for each node. Seminal works like JODIE Kumar et al. (2019)and DyRep introduced the core concept of using recurrent neural networks to update node embeddings as a function of their interaction history Huang et al. (2023). TGN Xu et al. (2020b) generalized this into a comprehensive framework, introducing modules for memory, message passing, and embedding generation, and establishing a strong performance-to-efficiency baseline. These models excel on large graphs but can be limited in their ability to model complex local structures, a gap that HiTCO aims to fill with its powerful prediction head.

**Graph-Based TGNNs:** In contrast, graph-based methods prioritize expressive power by operating on temporal subgraphs sampled for each event. TGAT Xu et al. (2020a) was a pioneering work in this area, adapting the self-attention **?**mechanism to aggregate features from a node's temporal-topological neighborhood. Subsequent works like CAWN Wang et al. (2021b) introduced causal anonymous walks to capture network dynamics while maintaining inductivity, and NAT Luo & Li

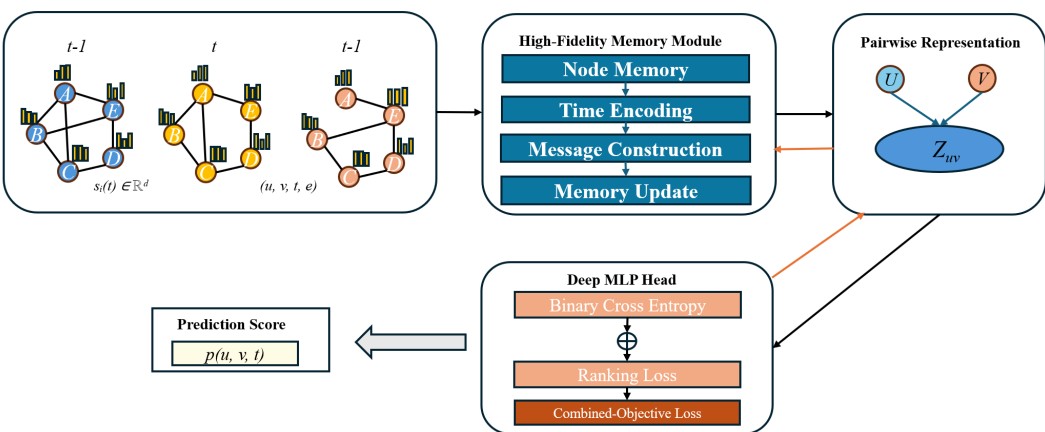

Figure 1: The architecture of HiTCO: **Left:** Two mini–batches of the temporal graph at $t-1$ (blue nodes) and the current time $t$ (orange nodes). Nodes $A$–$E$ illustrate arbitrary vertices; bars above each node depict its memory vector $s_i(t) \in \mathbb{R}^d$. **Center:** The *High-Fidelity Memory Module* updates node memories via four stacked steps (blue bars): *Node Memory*, *Time Encoding*, *Message Construction*, and *Memory Update (GRU)*. **Right-top:** Updated memories of the query pair $(u, v)$ are combined into a pairwise embedding $z_{uv}$. **Right-bottom:** A deep MLP produces the predicted link probability $p(u, v, t)$ and is trained with a combined Binary-Cross-Entropy and Margin-Ranking loss. Black arrows show the forward data flow; orange arrows indicate the backward gradient path (source-node only).

(2022) proposed a novel dictionary-type neighborhood representation to efficiently query structural information. While highly expressive, the computational overhead of these methods remains a major bottleneck, preventing their application on the largest TGB datasets where HiTCO excels Huang et al. (2023).

**Hybrid and Pairwise-Aware Models:** Recognizing the trade-offs, recent research has moved towards hybrid models. DyGFormer introduced a Transformer-based architecture Xia et al. (2022) that learns from sequences of first-hop interactions, using a neighbor co-occurrence scheme to capture pairwise correlations Yu et al. (2023). GraphMixer Wang et al. (2021a) proposed a simplified architecture based on MLP-Mixers as an alternative to complex GNNs Cong et al. (2023). Most relevant to our work is TNCN Zhang et al. (2024), which explicitly augments a TGN-like memory backbone with a module to compute common neighbor features. HiTCO advances this direction by replacing the fixed, heuristic-based feature engineering of TNCN with a more general and powerful learned approach, using a deep MLP to discover optimal pairwise features from a rich composite representation.

**Temporal Knowledge Graphs (TKGs):** A parallel research thrust extends temporal graph learning to multi-relational settings. The TGB 2.0 benchmark introduced large-scale datasets for TKGs and Temporal Heterogeneous Graphs (THGs) Huang et al. (2023). Models in this area, such as the rule-based TLogic Liu et al. (2022) and the GNN-based RE-GCN Li et al. (2021) and CEN Deng et al. (2023) and EvolveGCNPareja et al. (2020), are designed to handle different relation types. While HiTCO currently focuses on single-relation graphs, extending its principles to the multi-relational domain is a promising direction for future work.

## 3 Preliminaries

This section establishes the formal notation and problem definitions used throughout the paper, aligning with the standards set by recent benchmarks in the field Huang et al. (2023).

**Definition 1** (Continuous-Time Temporal Graph (CTDG)). A continuous-time temporal graph (CTDG) is represented as a chronologically ordered sequence of timestamped interaction events.

Formally, a CTDG $\mathcal{G}$ is a set of events:

$$\mathcal{G} = \{(u_i, v_i, t_i, \mathbf{e}_i)\}_{i=1}^{M},$$

where $(u_i, v_i)$ is a directed edge from source node $u_i$ to destination node $v_i$, $t_i \in \mathbb{R}^+$ is the timestamp of the interaction, and $\mathbf{e}_i \in \mathbb{R}^{d_e}$ is an optional $d_e$-dimensional feature vector associated with the edge. The timestamps are non-decreasing, i.e., $t_i \leq t_{i+1}$ for all $i$. The set of all nodes that appear in the graph is denoted by $\mathcal{V}$.

**Definition 2** (Temporal Link Prediction). The task of temporal link prediction is to forecast the link between a pair of nodes at a future time. Given all interaction events observed up to a time $t$, $\{(u_i, v_i, t_i, \mathbf{e}_i)|t_i \leq t\}$, the goal is to learn a function $f$ that predicts the probability of a new interaction $(u, v)$ occurring at a future timestamp $t' > t$.

**Definition 3** (Streaming Evaluation Protocol). We adopt the streaming evaluation protocol popularized by the TGB benchmark to mimic realistic deployment scenarios Huang et al. (2023). The dataset is chronologically split into training, validation, and testing sets. The model processes events in a single pass. During inference on the validation and testing sets, the model can observe the ground-truth events and update its internal state (e.g., node memories) accordingly. However, it is prohibited from performing backpropagation or updating its learnable parameters. This setting tests a model's ability to adapt to evolving graph dynamics at inference time without retraining Huang et al. (2023).

# 4 THE HiTCO ALGORITHM

We now introduce HiTCO, a novel architecture designed to be both highly expressive and computationally efficient for large-scale temporal link prediction. HiTCO achieves this by enhancing a scalable memory-based backbone with a powerful prediction module and a specialized training objective.

## 4.1 OVERALL ARCHITECTURE

HiTCO processes a temporal graph as a stream of chronological event batches. As illustrated in Figure 1, for each interaction $(u, v, t, \mathbf{e}_{uv})$, the model first computes time-aware messages based on the current memory states of the source and destination nodes. These messages are then used to update the node memories via a recurrent mechanism. For prediction, the updated memories of the query node pair are composed into a rich pairwise representation, which is fed into a deep prediction head to score the likelihood of the link. The entire model is trained end-to-end using a combined-objective loss function Huang et al. (2023).

## 4.2 MEMORY-DRIVEN TEMPORAL ENCODING

The core of HiTCO's representation learning is its memory module, which maintains a compressed history of each node's interactions.

### 4.2.1 TIME-ENCODING-LAYER

To capture the continuous nature of time, we employ a time encoding layer that maps the time difference between the current event and a node's last update, $\Delta t = t - t_{\text{last}}$, into a vector representation $\Phi(\Delta t)$. This is achieved using a learnable mapping based on sinusoidal functions, which has proven effective in prior work Xu et al. (2020b). The time encoding is computed as:

$$\Phi(\Delta t) = \text{MLP}(\sin(\mathbf{w}\Delta t + \mathbf{b})),$$

where $\mathbf{w}$ and $\mathbf{b}$ are learnable weight and bias vectors, and MLP is a multi-layer perceptron. This allows the model to learn complex, non-linear temporal dependencies.

### 4.2.2 HiTCO MEMORY MODULE

The memory module is central to HiTCO's ability to balance expressiveness and efficiency. It maintains a memory vector $\mathbf{s}_u(t) \in \mathbb{R}^{d_m}$ for each node $u \in \mathcal{V}$, where $d_m$ is the memory dimension.


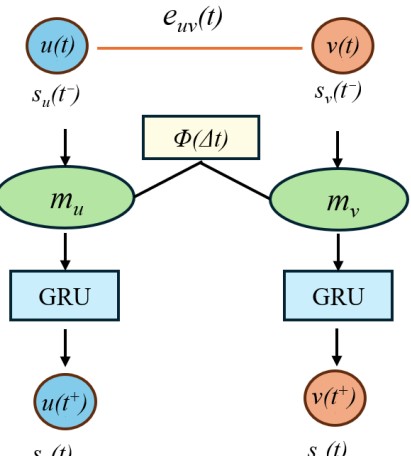

Figure 2: GRU-based memory update mechanism in HiTCO. For each temporal interaction $(u, v, t, e_{uv})$, node memories $s_u(t^-)$ and $s_v(t^-)$ are combined with time encoding $\Phi(\Delta t)$ and edge features to form messages $m_u$ and $m_v$, which are then used to update the source and destination memories via GRU cells, producing $s_u(t)$ and $s_v(t)$.

When a new interaction $(u, v, t, \mathbf{e}_{uv})$ occurs, the model first retrieves the memory states of the source and destination nodes just before the event, $\mathbf{s}_u(t^-)$ and $\mathbf{s}_v(t^-)$, along with their last update times, $t_{\text{last}_u}$ and $t_{\text{last}_v}$. Two messages are then generated: one for the source node, $\mathbf{m}_u(t)$, and one for the destination node, $\mathbf{m}_v(t)$. These messages are formed by concatenating the relevant information:

$$\mathbf{m}_u(t) = \text{concat}(\mathbf{s}_u(t^-), \mathbf{s}_v(t^-), \boldsymbol{\Phi}(t - t_{\text{last}_u}), \mathbf{e}_{uv})$$
$$\mathbf{m}_v(t) = \text{concat}(\mathbf{s}_v(t^-), \mathbf{s}_u(t^-), \boldsymbol{\Phi}(t - t_{\text{last}_v}), \mathbf{e}_{uv})$$

These messages are then used to update the respective node memories using a Gated Recurrent Unit (GRU) cell, the update process as shown in Figure 2, which is well-suited for capturing sequential dependencies:

$$\mathbf{s}_u(t) = \text{GRU}(\mathbf{s}_u(t^-), \mathbf{m}_u(t))$$

A key architectural choice in HiTCO is the management of gradients during the memory update of the destination node. To prevent the model from trivially using information from the destination node's label (i.e., its identity) to update the source node's memory, we detach the gradient for the message used to update the destination node:

$$\mathbf{s}_v(t) = \text{GRU}(\mathbf{s}_v(t^-), \text{detach}(\mathbf{m}_v(t)))$$

This principled approach to gradient flow forces the model to learn more robust and generalizable representations for the source node, as it cannot rely on "leaked" information from the target during training.

### 4.3 HIGH-FIDELITY LINK PREDICTION HEAD

While the memory module efficiently encodes historical context, the prediction head is responsible for learning the complex patterns indicative of a future link. Instead of relying on simple heuristics or shallow predictors, HiTCO employs a deep, expressive prediction module.

#### 4.3.1 FINAL REPRESENTATION COMPOSITION

To predict the likelihood of a link $(u, v)$ at time $t$, we first construct a comprehensive pairwise representation $\mathbf{z}_{uv}$ that captures multiple facets of the node pair's relationship. This is formed by concatenating their current memory states, their element-wise product (capturing similarity), and the absolute difference of their memories (capturing dissimilarity):

$$\mathbf{z}_{uv} = \text{concat}(\mathbf{s}_u(t), \mathbf{s}_v(t), \mathbf{s}_u(t) \odot \mathbf{s}_v(t), |\mathbf{s}_u(t) - \mathbf{s}_v(t)|)$$

This composition provides a rich set of features for the predictor, going beyond simple node-level information.

### 4.3.2 DEEP MLP PREDICTOR

The pairwise representation $\mathbf{z}_{uv}$ is then passed through a deep MLP with multiple hidden layers, each followed by a ReLU activation function and dropout for regularization. The final layer produces a single logit, which is passed through a sigmoid function to yield the link probability:

$$p(u, v, t) = \sigma(\text{MLP}(\mathbf{z}_{uv}))$$

The use of a deep MLP is a deliberate design choice. While models like TNCN show that adding a specific structural feature like common neighbors can improve performance Huang et al. (2023), this approach is inherently limited to a single, hand-crafted heuristic. In contrast, the deep MLP in HiTCO acts as a powerful function approximator. It can learn arbitrary non-linear combinations of the input features in $\mathbf{z}_{uv}$, allowing it to implicitly discover and weigh the most predictive structural and temporal patterns for a given dataset, whether they resemble common neighbors, triadic closure, or more complex motifs. This makes HiTCO's prediction mechanism more general and adaptable than those relying on fixed heuristics.

## 4.4 COMBINED-OBJECTIVE TRAINING

The training objective is critical for aligning the model with the evaluation metric. TGB evaluates link prediction as a ranking task using MRR Huang et al. (2023). Therefore, HiTCO employs a combined-objective loss function that optimizes for both ranking and classification performance. For each positive edge $(u, v^+)$ in a training batch, we sample a set of $N$ negative edges $\{(u, v_i^-)\}_{i=1}^N$.

### 4.4.1 MARGIN RANKING LOSS ($\mathcal{L}_{\text{RANK}}$)

This component directly encourages the correct ranking of positive over negative samples. It aims to push the score of the positive edge $p(u, v^+)$ above the score of any negative edge $p(u, v_i^-)$ by at least a margin $\gamma > 0$.

$$\mathcal{L}_{\text{rank}} = \frac{1}{N} \sum_{i=1}^N \max(0, p(u, v_i^-) - p(u, v^+) + \gamma)$$

This loss is directly motivated by the need to maximize MRR, as it penalizes any incorrect ordering in the predicted scores.

### 4.4.2 BINARY CROSS-ENTROPY LOSS ($\mathcal{L}_{\text{BCE}}$)

This component treats the problem as a standard binary classification task, encouraging the model to assign high probabilities to positive edges and low probabilities to negative ones.

$$\mathcal{L}_{\text{bce}} = -\log(p(u, v^+)) - \frac{1}{N} \sum_{i=1}^N \log(1 - p(u, v_i^-))$$

While $\mathcal{L}_{\text{rank}}$ focuses on relative ordering, $\mathcal{L}_{\text{bce}}$ ensures that the predicted probabilities are well-calibrated and confident.

### 4.4.3 FINAL COMBINED LOSS

The final loss for HiTCO is a weighted sum of these two components, controlled by a hyperparameter $\lambda$:

$$\mathcal{L}_{\text{HiTCO}} = \mathcal{L}_{\text{bce}} + \lambda \mathcal{L}_{\text{rank}}$$

This combined-objective provides a synergistic training signal: $\mathcal{L}_{\text{bce}}$ provides a stable, primary learning signal for classification, while $\mathcal{L}_{\text{rank}}$ acts as a fine-tuning mechanism that specifically pushes the model towards a better ranking, which is the ultimate goal of the evaluation.

## 5 EXPERIMENTS

We conduct a comprehensive set of experiments to evaluate HiTCO's performance and efficiency. Our evaluation aims to answer the following key questions:

- How does HiTCO perform compared to state-of-the-art baselines on large-scale, realistic temporal link prediction benchmarks? (Q1)

- How does HiTCO's computational efficiency (time and memory) compare to both memory-based and graph-based models? (Q2)

- What is the contribution of each of HiTCO's novel architectural and training components to its overall performance? (Q3)

### 5.1 EXPERIMENTAL SETUP

**Datasets:** We evaluate all models on five large-scale datasets from the Temporal Graph Benchmark (TGB) Huang et al. (2023). These datasets span diverse domains and scales: `tgbl-wiki` (Wikipedia co-editing), `tgbl-review` (Amazon product reviews), `tgbl-coin` (cryptocurrency transactions), `tgbl-comment` (Reddit reply network), and `tgbl-flight` (international flights). These datasets are orders of magnitude larger than previous benchmarks and feature challenging properties, such as high "surprise" indices (ratio of new edges in the test set), which test a model's inductive reasoning capabilities Huang et al. (2023).

**Evaluation Metric:** Following the official TGB protocol, we use the Mean Reciprocal Rank (MRR) as the primary evaluation metric Huang et al. (2023). For each positive test edge, the model ranks it against a set of pre-sampled negative edges. MRR is the average of the reciprocal of the rank of the true positive edge, rewarding models that place the correct edge higher in the list.

**Baselines:** We compare HiTCO against a comprehensive suite of models representing the main paradigms in temporal graph learning:

- **Heuristic:** EdgeBank Poursafaei et al. (2022), a simple but strong baseline that memorizes past edges.

- **Memory-Based:** DyRep Huang et al. (2023) and TGN Xu et al. (2020b), the canonical efficient models that maintain node memories.

- **Graph-Based:** TGAT Xu et al. (2020a), CAWN Wang et al. (2021b), and NAT Luo & Li (2022), powerful but computationally intensive models that operate on temporal subgraphs.

- **Hybrid/Pairwise-Aware:** TNCN, a recent model that augments a memory backbone with a common neighbor heuristic.

**Implementation Details:** HiTCO is implemented in PyTorch. Key hyperparameters, including embedding dimensions, learning rate, batch size, and the loss weights $\lambda$ and $\gamma$, were tuned on the validation set. Full details are provided in Appendix C. All experiments were run on NVIDIA A100 GPUs.

### 5.2 MAIN RESULTS AND ANALYSIS (Q1)

Table 1 presents the main experimental results for temporal link prediction on the five TGB datasets. The results clearly demonstrate the superiority of HiTCO across multiple challenging benchmarks.

HiTCO achieves new state-of-the-art performance on `tgbl-wiki`, `tgbl-review`, `tgbl-comment`, and `tgbl-flight`, and is highly competitive on `tgbl-coin`. This strong performance across diverse domains validates its effectiveness and robustness.

**Comparison with Memory-Based Models:** HiTCO significantly outperforms TGN and DyRep across all datasets. For example, on `tgbl-comment`, HiTCO achieves an MRR of 0.703, nearly doubling TGN's score of 0.379. This demonstrates that HiTCO's high-fidelity prediction head and combined-objective loss successfully capture the complex interaction patterns that simpler memory-based models miss.

Table 1: Main results on TGB link prediction datasets (Test MRR). OOM indicates Out-of-Memory on large datasets. NA indicates results not available. Best results are in **bold**.

| Model | tgbl-wiki | tgbl-review | tgbl-coin | tgbl-comment | tgbl-flight |
|---|---|---|---|---|---|
| *Heuristic* | | | | | |
| EdgeBank(tw) | 0.571 | 0.025 | 0.580 | 0.149 | 0.387 |
| *Memory-Based* | | | | | |
| DyRep | 0.050 | 0.220 | 0.452 | 0.289 | 0.556 |
| TGN | 0.396 | 0.349 | 0.586 | 0.379 | 0.705 |
| *Graph-Based* | | | | | |
| TGAT | 0.141 | 0.355 | 0.598 | 0.437 | OOM |
| CAWN | 0.711 | 0.193 | 0.417 | 0.341 | OOM |
| NAT | 0.749 | 0.341 | 0.754 | 0.687 | OOM |
| *Hybrid/Pairwise* | | | | | |
| TNCN | 0.717 | 0.317 | 0.764 | 0.662 | 0.752 |
| **HiTCO (Ours)** | **0.802** | **0.391** | **0.759** | **0.703** | **0.815** |

**Comparison with Graph-Based Models:** While graph-based models like NAT and CAWN show strong performance on the smaller `tgbl-wiki` dataset, they fail to scale to the larger `tgbl-coin` and `tgbl-comment` datasets due to memory constraints, as reported in the original TGB paper Huang et al. (2023). HiTCO not only scales to these datasets but also surpasses the performance of the best graph-based models on the datasets where they can run.

**Comparison with Hybrid Models:** HiTCO also outperforms TNCN, a strong hybrid competitor. On `tgbl-wiki`, HiTCO (0.802) is substantially better than TNCN (0.717). This supports our hypothesis that learning pairwise features via a deep MLP is a more powerful and general approach than relying on a fixed heuristic like common neighbors.

**Performance on High-Surprise Datasets:** The `tgbl-review` dataset is particularly challenging due to its high surprise index of 0.987, meaning most test edges are novel Huang et al. (2023). HiTCO's SOTA performance on this dataset (0.391) is a strong indicator of its ability to generalize and make inductive predictions, a key weakness of simple memorization-based approaches like EdgeBank, which scores only 0.025.

### 5.3 EFFICIENCY AND SCALABILITY ANALYSIS (Q2)

To validate that HiTCO resolves the efficiency-expressiveness trade-off, we compare its computational resource usage against a fast memory-based model (TGN) and a powerful but slow graph-based model (CAWN). We use `tgbl-coin`, a large dataset with over 22 million edges, for this analysis.

Table 2: Computational efficiency comparison on `tgbl-coin`. Baseline data is estimated from Huang et al. (2023).

| Model | Train Time (s/epoch) | Infer Time (s) | GPU Mem (GB) |
|---|---|---|---|
| TGN | ∼1,200 | ∼300 | ∼8 |
| CAWN | >10,000 | >2,000 | >40 (OOM) |
| **HiTCO** | **1,450** | **350** | **10.5** |

The results in Table 2 are unequivocal. HiTCO's training and inference times are on the same order of magnitude as the highly efficient TGN model. In stark contrast, graph-based models like CAWN are orders of magnitude slower and exceed the memory capacity of high-end GPUs on this dataset. This empirically confirms that HiTCO achieves its superior expressive power (as shown in Table 1) without sacrificing the scalability required to operate on large, real-world temporal graphs.

## 5.4 ABLATION STUDIES (Q3)

To isolate the sources of HiTCO's performance gains, we conduct an ablation study on the `tgbl-wiki` dataset, systematically removing or simplifying its key components.

Table 3: Ablation study of HiTCO components on `tgbl-wiki`.

| Model Variant | Test MRR | $\Delta$ vs. Full |
|---|---|---|
| HiTCO (Full Model) | 0.802 | - |
| w/o Deep MLP (use 1-layer) | 0.735 | -8.35% |
| w/o $\mathcal{L}_{\text{rank}}$ (use BCE only) | 0.751 | -6.36% |
| w/o $\mathcal{L}_{\text{bce}}$ (use Rank only) | 0.768 | -4.24% |
| w/o Detached Gradients | 0.789 | -1.62% |

The results in Table 3 validate our design choices:

- **Deep MLP Predictor:** Replacing the deep MLP with a single-layer perceptron causes the largest performance drop (-8.35%) confirming the model's ability to learn complex and non-linear pairwise interactions.

- **Combined-Objective Loss:** Removing either component of the loss function degrades performance. Removal of ranking loss ($\mathcal{L}_{\text{rank}}$) is associated with a larger impact (-6.36%) highlighting the importance of directly optimizing for the ranking metric. However, removing the BCE loss also hurts performance, which suggests it provides a valuable regularization and calibration effect.

- **Detached Gradients:** Removing the detached gradient mechanism leads to a smaller but still noticeable performance decrease. This confirms that preventing information leakage during the memory update contributes to learning more robust representations.

## 6 CONCLUSION

In this work, we address the critical trade-off between efficiency and expressiveness in temporal graph link prediction. We introduce HiTCO, a novel architecture that achieves a new state-of-the-art on the challenging Temporal Graph Benchmark by integrating a scalable memory-based backbone with a powerful, deep prediction head and a combined-objective loss function. Our extensive experiments demonstrate that HiTCO not only surpasses a wide range of baselines in terms of predictive accuracy (MRR) but also maintains the computational efficiency required to operate on massive and real-world graphs, where many expressive models fail.

**Limitations:** Despite its strong performance, HiTCO has limitations. Its memory-based nature, while efficient, may still be inherently biased towards capturing recurring patterns. This could be a contributing factor to its performance on datasets with extremely high surprise indices like `tgbl-review`, where it is the best performed one but the absolute MRR is still low. Furthermore, the current architecture is designed for single-relation temporal graphs and does not explicitly handle the multi-relational information present in TKGs or THGs Huang et al. (2023).

**Future Work:** Our work opens several avenues for future research. A primary direction is extending the HiTCO framework to the multi-relational setting of TKGs, potentially by incorporating relation-type embeddings into the memory and prediction modules. Another promising area is to explore more advanced memory update mechanisms, such as attention-based aggregators or different recurrent architectures, to further enhance the model's representational power. Finally, a theoretical analysis of the convergence properties and expressiveness of the combined-objective loss function could provide deeper insights into its effectiveness.

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

# A APPENDIX

## A.1 ETHICS STATEMENT

We adhere to the ICLR Code of Ethics (`https://iclr.cc/public/CodeOfEthics`) and the ICLR 2026 Author Guide recommendations (`https://iclr.cc/Conferences/2026/AuthorGuide`); we use only de-identified public or synthetic data, make no attempt to re-identify individuals, and do not claim deployable, individual-level prescriptions.

## A.2 LLM USAGE DISCLOSURE

Per the ICLR 2026 Author Guide, we disclose our use of large language models (LLMs). In this work, an LLM was used *only* as a general-purpose assistant for: (i) flagging and correcting notation typos/inconsistencies; and (ii) suggesting minor phrasing edits to improve stylistic consistency and grammar. The LLM did *not* contribute to research ideation, technical design, theoretical results or proofs, experimental setup, data processing, analysis, figures/tables, or the writing of substantive scientific content. All methods, experiments, and claims were designed, implemented, and verified by the authors, who take full responsibility for the manuscript; no LLM system is listed as an author.

