# OpenReview forum: "HiTCO: High-Fidelity Memory and Combined-Objective Training for Dynamic Link Prediction"
_ICLR.cc/2026/Conference — ICLR 2026 Conference Withdrawn Submission_

### Official Review · Reviewer_PsdD · 2025-10-24

**Soundness:** 2
**Presentation:** 3
**Contribution:** 2
**Rating:** 2
**Confidence:** 3

**Summary:**

This paper introduces HiTCO, a temporal graph link prediction model designed to balance the trade-off between efficiency and expressiveness. HiTCO extends the memory-based paradigm (e.g., TGN) by combining three innovations:a high-fidelity GRU-based memory module with careful gradient management, a deep MLP prediction head that learns complex pairwise node interactions, and a combined-objective loss integrating binary cross-entropy (classification confidence) and margin-ranking loss (ranking accuracy).

The paper positions HiTCO as a principled alternative to hybrid models that rely on handcrafted heuristics such as common neighbors. Extensive experiments on five large-scale TGB datasets show consistent state-of-the-art performance, significantly outperforming memory-based, graph-based, and hybrid baselines. Ablation studies further support the contribution of each architectural component.

Overall, the paper offers a strong engineering improvement and practical contribution to scalable temporal graph modeling, though its conceptual novelty over recent hybrid methods remains modest.

**Strengths:**

* The paper’s design choices—detaching gradients for destination nodes, using pairwise composition features, and employing a deep MLP predictor—are well-motivated and systematically justified through ablation studies.
* The integration of margin ranking and BCE losses aligns the training objective with evaluation metrics. This combination is both intuitive and empirically validated as a key factor in achieving better ranking performance.
* The paper is clearly structured, follows standard temporal graph notation, and provides thorough comparisons with appropriate baselines. Implementation details and datasets align with community benchmarks, enhancing reproducibility.

**Weaknesses:**

* Novelty and conceptual contribution：While the empirical performance is strong, the core methodological innovation is relatively incremental. HiTCO primarily refines the existing memory-based framework (TGN/TNCN) rather than introducing a fundamentally new modeling paradigm. The deep MLP predictor replaces heuristic structural encodings, but this substitution—though effective—is conceptually straightforward.
* Theoretical justification and analysis：The paper lacks theoretical insights into why the combined-objective loss improves generalization or why gradient detachment leads to more robust representations. A formal analysis of convergence, expressiveness, or complexity would strengthen the paper’s scientific depth.
* Experimental scope and fairness：Although results on TGB are comprehensive, the comparison set could be expanded to include more recent transformer-based temporal models (e.g., DyGFormer[1], GraphMixer[2]) or contrastive frameworks. Moreover, the ablations are limited to one dataset (tgbl-wiki); it would be preferable to validate across multiple datasets to ensure robustness.
* Presentation and discussion：While the paper is generally well written, the discussion of limitations is brief and somewhat superficial. For instance, the potential bias of memory mechanisms toward recurring patterns could be explored in more depth, and future directions could be more concretely articulated (e.g., toward multi-relation extensions or attention-based memories).

[1] Yu, Le, et al. "Towards better dynamic graph learning: New architecture and unified library." *Advances in Neural Information Processing Systems* 36 (2023): 67686-67700.

[2] Cong, Weilin, et al. "Do we really need complicated model architectures for temporal networks?." *arXiv preprint arXiv:2302.11636* (2023).

**Questions:**

*  Conceptual novelty：Could the authors clarify the fundamental innovation of HiTCO beyond improved integration of existing memory-based and hybrid frameworks?
*  Theoretical grounding：What theoretical or empirical evidence supports the effectiveness of the combined BCE + ranking objective and the gradient detachment strategy?
*  Experimental scope and fairness：Although results on TGB are comprehensive, the comparison set could be expanded to include more recent transformer-based temporal models (e.g., DyGFormer[1], GraphMixer[2]) or contrastive frameworks. Moreover, the ablations are limited to one dataset (tgbl-wiki); it would be preferable to validate across multiple datasets to ensure robustness.

*  Scalability assessment：How does the deeper MLP head affect inference latency and memory usage on large-scale or streaming graphs compared with TGN？
*  Generalization and extension：How might HiTCO be adapted to multi-relational or heterogeneous temporal graphs while preserving its computational efficiency?

[1] Yu, Le, et al. "Towards better dynamic graph learning: New architecture and unified library." *Advances in Neural Information Processing Systems* 36 (2023): 67686-67700.

[2] Cong, Weilin, et al. "Do we really need complicated model architectures for temporal networks?." *arXiv preprint arXiv:2302.11636* (2023).

---

### Official Review · Reviewer_fsqi · 2025-10-26

**Soundness:** 2
**Presentation:** 1
**Contribution:** 2
**Rating:** 2
**Confidence:** 4

**Summary:**

This work considers the temporal link prediction task on continuous-time dynamic graphs (CTDGs). Aiming to achieve significant trade-off between the computational efficiency of memory-based models and the expressive power of graph-based approaches, a new HiTCO (High-Fidelity Temporal Representation with Combined-Objective Training) method was introduced to leverage advantages of both types of methods. Experiments on large-scale dynamic graph benchmarks with the streaming evaluation protocol (i.e., TGB) preliminarily showed HiTCO's effectiveness.

**Strengths:**

**S1**. The proposed method was evaluated on large-scale temporal graph benchmarks with the streaming evaluation protocol (i.e., TGB).

**S2**. There are discussions about the limitations of this work and possible future research directions at the end of this paper.

**Weaknesses:**

**W1**. The overall presentation of this paper needs significant improvement.

The title of the submitted version is 'FORMATTING INSTRUCTIONS FOR ICLR 2026 CONFERENCE SUBMISSIONS', which is the default title of the submission template and is different from that shown in the submission system.

In Section 1, there are lengthy discussions about the advantages and disadvantages of existing techniques (e.g., memory-based, graph-based, and hybrid methods), which are hard to understand. It is recommended to summarize their key conclusions in a table.

The overall English presentation of this paper is hard to read, which needs further polishing.

This paper fails to fully utilize the page limit of submission (i.e., 9 pages of this main paper), where some more details (see **W2**-**W5**) can be added.


***
**W2**. There are many nuclear statements with weak motivations, which need further clarification.

In Section 1, it was claimed that 'memory-based methods' reliance on node-level representations and simple aggregation functions limits their capacity to capture complex high-order structural patterns', but why? It is suggested to add some toy running examples (e.g., a small temporal graphs with few nodes) to further interpret this statement.

In Section 1, it was claimed that graph-based methods can achieve high expressiveness. However, it is unclear how to measure the expressiveness (e.g., using what metrics) and why they can achieve high expressiveness.

In Section 1, it was claimed that graph-based methods suffer from high computational and memory costs. However, there are no discussions or summary to quantitatively show their time and space complexities.

It was highlighted that HiTCO benefits from a high-fidelity GRU-based memory module. Nevertheless, it is unclear how to define and measure a memory module is high-fidelity.

In Section 3 (Preliminaries), only edge attributes were mentioned. For most related methods, node attributes are also their optional inputs (e.g., used for the initialization of node-level embeddings). It is unclear whether this work assumes node attributes are available.

In Section 4.3.1., ${\bf{z}}_{uv}$ was defined based on element-wise product and absolute difference between ${\bf{s}}_u (t)$ and ${\bf{s}}_v (t)$. It is unclear what are motivations of considering such two operations. There seem no further ablation studies to verify their effectiveness.

There is no pseudo-code to summarize the overall training and inference procedures of HiTCO. As a results, some details of the proposed methods remain unclear (e.g., how to initialize the memory).

***
**W3**. While better space- and time-efficiency are the highlighted advantages of HiTCO, there is no any formal analysis about its time and space complexities as well as comparison with those of other SOTA baseline approaches.


***
**W4**. Novelty of this work seems to be incremental and limited. In particular, simply replacing a heuristic feature feature extraction module (i.e., that of TNCN) with MLP and combing two existing training losses cannot be considered as significant original contributions.


***
**W5**. Experiments are simple, which cannot fully validate the effectiveness of HiTCO.

Most of the baselines are too old. Only 1 baseline was published in 2024 but now is approaching the end of 2025.

In Table 2, the time and memory consumption of HiTCO were only compared with those of TGN and CAWN, while results of other baselines (e.g., EdgeBank, DyRep, TGAT, NAT, and TNCN) were not reported.

There is no parameter analysis to investigate the effects of some major hyper-parameters (i.e.,, $\lambda$ and $\gamma$) in the model training.


***
**W6**. References of this paper are too old. As the end of 2025 is approaching, there is only 1 reference published in 2024 and there are no any references published in 2025.


***
**W7**. This work did not anonymously provide its code and data to ensure the reproducibility of experiments.

**Questions:**

See **W1**-**W7**.

---

### Official Review · Reviewer_ryz4 · 2025-10-26

**Soundness:** 1
**Presentation:** 1
**Contribution:** 1
**Rating:** 2
**Confidence:** 4

**Summary:**

This paper modifies the graph-based embedding module of TGN into a pairwise schema, where edge embedding is computed merely based on the memories of two end nodes. Besides, this paper changes the binary cross-entropy loss into a combination of BCE and rank loss.

**Strengths:**

This studied problem of efficient and expressive dynamic graph neural networks is interesting and practical.

**Weaknesses:**

- The division of existing works is confusing. There lacks a clear boundary that can distinguish different types of methods.  Take the memory-based TGNNs as an example, if the authors frame it as “maintaining a memory state for each node”, then why is the NAT that maintains a memory state for each node classified into Graph-based TGNNs? The Hybrid and Pairwise-Aware Models are more convincing. What characteristics make them different from the Memory-based and Graph-based Methods? The reviewer suggests that the authors reorganize existing methods by a clear boundary.
- The experimental results are unconvincing. Strong baselines of the TGB Leaderboard, like DyGFormer and TPNet, are all ignored in Table 1. Moreover, Table 2 only compares two baselines on one dataset for computational efficiency, which is insufficient. The content of “Baseline data is estimated from Huang et al. (2023).” in the table header is also confusing. What does the “estimated form” mean?
- There are lots of typos in this paper. For example, the title of this paper is “FORMATTING INSTRUCTIONS FOR ICLR 2026 CONFERENCE SUBMISSION”. The citation of the self-attention in the related work section is a “?”.

**Questions:**

Please see the Weaknesses.

---

### Official Review · Reviewer_XAhH · 2025-11-01

**Soundness:** 2
**Presentation:** 3
**Contribution:** 2
**Rating:** 4
**Confidence:** 4

**Summary:**

This paper proposes a link prediction method for temporal graphs that aims to combine the advantages of both memory-based and GNN-based approaches. The goal is to achieve efficient training while still capturing structural pairwise information. Specifically, the model employs a GRU-based module to learn time-aware memory representations and a deep MLP to model pairwise interactions between nodes. Experimental results demonstrate superior performance compared with a variety of baselines.

**Strengths:**

1. The idea of combining the efficiency of memory-based models with the structural awareness of GNN-based approaches is interesting and relevant to temporal graph learning.

2. The paper also explores integrating both classification and ranking losses, which makes the objective more consistent with the ranking nature of the link prediction task.

**Weaknesses:**

1. The notion of "memory" is somewhat ambiguous. It appears to refer both to the model’s learned memory representations and to the reduced hardware memory consumption. It would improve clarity to clearly distinguish between these two meanings.

2. It is not entirely straightforward that the deep MLP can effectively capture structural information. Many prior link prediction methods explicitly encode pairwise or path-based information (e.g., [1], [2]). Providing additional analysis or qualitative case studies would help justify this claim.

[1] Neural Common Neighbor with Completion for Link Prediction

[2] Neural Bellman-Ford Networks: A General Graph Neural Network Framework for Link Prediction

3. The paper could include a more detailed analysis of the joint loss, particularly the effect of the weighting parameter $\lambda$, to better illustrate how the classification and ranking components interact and influence model performance.

**Questions:**

please refer to the weakness

---

### Note · Authors · 2025-11-28

I have read and agree with the venue's withdrawal policy on behalf of myself and my co-authors.